# Urolithins: A Prospective Alternative against Brain Aging

**DOI:** 10.3390/nu15183884

**Published:** 2023-09-06

**Authors:** Lei An, Qiu Lu, Ke Wang, Yousheng Wang

**Affiliations:** 1Key Laboratory of Geriatric Nutrition and Health, Ministry of Education, Beijing Technology and Business University, Beijing 100048, China; andijia1@126.com (L.A.); echo6939@163.com (Q.L.); wangke9509@163.com (K.W.); 2College of Light Industry and Food Engineering, Guangxi University, Nanning 530004, China; 3Rizhao Huawei Institute of Comprehensive Health Industries, Shandong Keepfit Biotech. Co., Ltd., Rizhao 276800, China

**Keywords:** urolithin, brain, aging, neurodegenerative diseases, microbial-derived, metabolites

## Abstract

The impact of host–microbiome interactions on cognitive health and disease has received increasing attention. Microbial-derived metabolites produced in the gut are one of crucial mechanisms of the gut–brain axis interaction, showing attractive perspectives. Urolithins (Uros) are gut microbial-derived metabolites of ellagitannins and ellagic acid, whose biotransformation varies considerably between individuals and decreases greatly with age. Recently, accumulating evidence has suggested that Uros may have specific advantages in preventing brain aging including favorable blood–brain barrier permeability, selective brain distribution, and increasingly supporting data from preclinical and clinical studies. However, the usability of Uros in diagnosis, prevention, and treatment of neurodegenerative diseases remains elusive. In this review, we aim to present the comprehensive achievements of Uros in age-related brain dysfunctions and neurodegenerative diseases and discuss their prospects and knowledge gaps as functional food, drugs, or biomarkers against brain aging.

## 1. Introduction

To date, the population of dementia is approximately 45 million worldwide (this number will double if mild cognitive impairment (MCI) is included), and this population is expected to triple (approximately 130 million) by 2050 [1]. As the population of older adults is increasing, interest in the aging brain and the quest for “healthy brain aging” is growing. Modern medicine and nutrition evidence has confirmed that polyphenol consumption is an effective strategy in delaying brain aging [2]. However, the poor bioavailability of dietary polyphenols in vivo is contradictory to their health benefits [3,4]. Recently, there has been increasing evidence suggesting that the health benefits of polyphenols are mainly based on: (1) Polyphenols being either directly absorbed or in most cases converted into bioactive metabolites by gut microbiota and then get to the target tissues; (2) Polyphenols reshaping the gut microbiome during their bidirectional interaction. In addition, microbial-derived metabolites of polyphenols can be used as biomarkers to predict individual health status or even the efficacy of some drugs in vivo [5]. Over the past decade, research on gut microbiota and the gut–brain axis has brought increasingly new insights into humans [6]. Several pathways have been confirmed in the communication of the gut microbiota and the brain: (1) neuroendocrinology: the gut cells secrete a large amount of the signaling molecules in neuroendocrinology which can be influenced by gut microbiota; (2) neuroimmunity: the GI tract itself is the largest immune organ to stress response and the gut microbiota can promote immune cells to produce cytokines that can affect the brain; (3) neurotransmitters: the gut microbiota can affect the neurotransmitter secretion directly (several neurotransmitters, including acetylcholine, dopamine, noradrenaline, and serotonin can be synthesized by gut bacteria) and indirectly; and (4) microbial-derived metabolites: neuroactive metabolites mediate bidirectional interactions between the gut, gut microbiome, and the brain to modulate host neurophysiology and behavior [7,8,9].

As many of the host–microorganism interactions pertaining to human health are mediated by metabolites, interest in the microbial-derived metabolites has been increasing [10,11,12]. Among these active metabolites, urolithins (Uros), the gut bacterial metabolites of ellagitannins (ETs) and ellagic acid (EA), have shown to be beneficial in delaying many age-related diseases such as cancer (especially prostate, breast, and colorectal cancers), cardiovascular diseases, and chronic metabolic disorders (diabetes and hyperuricemia) [13,14,15,16,17,18]. Recently, growing evidence has supported the multiple health benefits of Uros against brain aging and the therapeutic potential for neurodegenerative diseases [17,19,20,21,22]. Moreover, Uros have been proposed as biomarkers of gut dysbiosis and disease stage in Parkinson’s patients [23]. Herein, we aim to review the recent achievements of Uros in brain aging and neurodegenerative diseases, with a particular emphasis on their therapeutic targets and mechanisms and discuss their prospects and knowledge gaps.

## 2. Overview and Advantage of Uros

Uros are gut metabolites derived from ETs and EA, which are richly available in many fruits (pomegranates, berries), nuts (walnuts), wood-aged wine, and some medicinal plants (galla chinensis, chebulae fructus, and seabuckthorn leaf) [12]. ETs and EA have very poor bioavailability in vivo and are converted into Uros by intestinal microbes in the colon, which are believed to be responsible for their biological activities [24,25]. ETs are a group of important dietary polyphenols and hydrolysable tannins that share a common core, hexahydroxydiphenoyl (HHDP), and differ in the number of monomer residues. ETs are mainly hydrolyzed to EA in the upper GI tract (stomach and small intestine), and most of the EA is converted to Uros in the lower GI tract (mainly in the colon) [25,26,27]. Uros are 6H-dibenzo [b, d] pyran-6-one derivatives which differ in their hydroxyl groups. The family includes the main end metabolites, dihydroxy derivatives UroA, Iso-UroA and monohydroxy derivative UroB, and other intermediates UroC, UroD, UroE, UroM-5, UroM-6, and UroM-7. Overall, UroA is the main metabolite produced in humans, which demonstrates the highest concentrations in plasma and urine, and remains high for days after the consumption of ET-rich food [28,29,30].

The bioconversion of Uros from ETs and EA varies considerably between individuals. Some individuals can produce high plasma concentrations of Uros (high Uro producers), while others cannot produce significant levels of Uros (low Uro producers). Additionally, Uro producers can also be classified into “metabotype A” (only high concentration of UroA is produced), and “metabotype B” (more Iso-UroA and/or UroB in comparison to UroA is produced) [31]. Therefore, the consumption of dietary ETs will produce different health benefits in high or low Uro producers, and those with different metabolic phenotypes. The different capacities for excreting Uros are mainly attributed to the variability of gut microbiota ecology, and vary with age, health status, dietary intake, obesity, and digestive organ surgery [32,33]. Notably, age is a key factor that affects the bioconversion of Uros. The bioconversion of Uros significantly decreases with aging. According to clinical observations, approximately 10% of individuals (aged 5–90) are non-Uro producers, and only 40% of elderly people (>60) can produce meaningful levels of Uros from dietary precursors [34]. Therefore, Uro supplementation may be a good alternative for certain individuals (e.g., elderly people) to meet the required healthy Uro level [35].

In comparison to precursors and live bacteria, Uros possess clear chemical structure, good bioavailability, and high safety in animals and humans [36,37]. Recently, two multi-center clinical trials showed that long-term (4-month) oral administration of UroA is safe and well tolerated [38,39]. Moreover, UroA is considered as GRAS (generally recognized as safe) for its use in food products by the Food and Drug Administration (FDA) [40]. Notably, Uros can also pass through the blood–brain barrier (BBB) and distribute in a brain-targeted manner after absorption [17,41,42], which may greatly facilitate their activities in the central nervous system (CNS), while other ET derivatives cannot cross the BBB. Additionally, as increasing evidence supporting the involvement of the gut metabolites in human health and diseases, metabolite-based treatment has been considered as a novel and promising therapeutic approach. It can overcome limitations and deficiencies of microbiome-based treatment (probiotics and prebiotics supplementation or fecal microbiome transplantation), such as colonization resistance and inter-individual variation in microbial composition [43]. Microbial-derived metabolites may be able to provide an improved efficacy (safety, stability, and individual variation) by exerting a beneficial host effect downstream of the microbiome. Collectively, these findings highlight the advantages of Uros in developing into functional food or drugs. 

Many studies have investigated the biotransformations of ETs and EA to Uros. ETs are first converted into EA, facilitated by physiological pH and/or microbial enzymes such as tannin–hydrolase and lactonase. EA is then catalyzed by several microbial enzymes, including lactonase, decarboxylase, dehydroxylases, methyl esterases, and hydrolases, to produce Uros in the colon [25]. The metabolic pathways associated with these enzymes have been elucidated. Recently, several Uro-producing gut bacteria have been identified that can facilitate the conversion of EA to Uros in vitro [44,45,46]. Therefore, in addition to the chemical synthesis of Uros, the biotransformation of Uros with bacteria or enzymes in vitro in the near future can be expected [28,47]. 

As previously described, the consumption of dietary polyphenols is highly correlated with human health and the risk of neurodegenerative diseases [12]. Dynamic two-way communications between the gut, gut microbiome, and the CNS have been considered an increasingly vital factor for cognitive health and disease. Novel perspectives suggest that microbial-derived metabolites of polyphenols can be used as biomarkers to predict the risk of neurodegenerative diseases [48]. Therefore, in addition to the potential as functional food or drugs, Uros may be potential biomarkers of age-related cognitive decline and related disorders, which is mainly based on the following: (1) As bioactive metabolites, they exert multiple health benefits against brain aging and related diseases (which we will discuss below in detail); (2) the level of Uros in baseline is an indicator of personalized nutrition status (the consumption of dietary polyphenols in individuals, particularly for ETs and EA consumption), which is highly correlated with the risk of neurodegenerative diseases; (3) the level of Uros after consuming ET- or EA-rich food is an indicator of individual metabotypes associated with specific gut microbial ecologies (gut microbiota composition and functionality), which is tightly involved in neurodegenerative diseases. A recent study based on 52 patients and 117 healthy individuals showed that gut dysbiosis occurred during the onset and progression of Parkinson’s disease (PD) and the concentration of Uros in urine was highly associated with the severity of gut dysbiosis and PD in elderly people [23].

## 3. Preclinical and Clinical Studies

As UroA and UroB are the most abundant final metabolites in vivo, they are the most widely studied Uros across species, particularly UroA. Accumulating preclinical studies have demonstrated the beneficial effects of UroA and UroB on age-related brain dysfunctions. The cell studies in vitro are summarized in Table 1. Although these studies all demonstrated the neuroprotective effect of Uros in aging conditions such as oxidative stress, inflammation, and high concentrations of glucose, the treatment conditions in different laboratories and the sensitivity of different cells to Uros varied considerably. The animal studies in vivo are summarized in Table 2. Uros administration has been shown significantly improve cognition, learning, and memory in aged animals and animal models of neurodegenerative diseases.

It has been reported that ET- or EA-rich food consumption improve cognition and memory in the elderly (summarized in Table 3), whereas the effect of Uros supplementation in the elderly is still unknown. Two sequential randomized, double-blind, placebo-controlled trials in middle-aged and older adults (aged 50–75 years) with MCI showed that drinking 8 ounces (237 mL) of pomegranate juice every day for short-term 4 weeks or long-term 12 months significantly improved performance in the Brief Visuospatial Memory Test–Revised and Buschke Selective Reminding Test, and increased functional brain activation during verbal and visual memory tasks (ClinicalTrials.gov Identifier: NCT02093130) [42,49]. Moreover, cognition and memory improvement by pomegranate juice consumption is in line with the concentration of UroA–glucuronide in plasma and the plasma Trolox-equivalent antioxidant capacity. Additionally, berry and walnut consumption are related to better cognitive function and memory performance in the elderly with normal cognition or MCI [50,51,52,53]. Recently, several clinical trials have shown that UroA administration improves skeletal muscle function and mitochondrial health in elderly and middle-aged adults (ClinicalTrials.gov Identifier: NCT02655393, NCT03464500, and NCT03283462) [37,38,39]. In summary, evidence supporting the role of Uros in preventing the degeneration of brain during aging is accumulating, whereas clinical study in humans is lacking and needs further investigation.

**Table 1 nutrients-15-03884-t001:** Preclinical studies of Uros on brain aging in vitro cell models.

Uros	Cells	Pharmacological or Genetic Interventions	Treatment (Dosage and Time)	Effects	Findings	Refs.
UroA	Neuro-2a cells	H_2_O_2_ (250 μM) for 45 min	0.5 μM, 1 μM, 2 μM, 4 μM pretreatment for 24 h	antioxidation	↑ cells viability, ↓ MAO-A and Tyrosinase, ↑ free radical (O^2−^ and DPPH), ↓ ROS, lipid peroxidation, ↑ peroxiredoxins expression, ↑ CAT, SOD, GR, GSH-Px,	[54]
UroA	PC12 cells	H_2_O_2_ (100 μM) for 2 h	10 μg/mL, 30 μg/mL, and 50 μg/mL pretreatment for 24 h	antioxidation	↑ cells viability, ↓ LDH release, ↓ apoptosis, ↓ caspase 3 and Bcl-2	[17]
UroA	SK-N-MC cells	H_2_O_2_ (300 μM) for 18 h	1.25 μM, 2.5 μM, and 5 μM pretreatment for 6 h	antioxidation	↑ cells viability, ↓ apoptosis, ↓ ROS, Bax/Bcl-2, PARP, cytochrome c, caspase 3/9, p38 MAPK	[55]
UroA	SH-SY5Y cells	H_2_O_2_ (100 μM) for 24 h	10 μM treatment for 2 h, 6 h or 24 h	antioxidation	↑ REDOX activity, ↓ cytotoxicity, ↓ ROS, ↓ apoptosis, ↓ caspase 3/8 and 9	[20]
UroA	SH-SY5Y cells	H_2_O_2_(200 μM) for 6 h	5 μM, 7.5 μM, 10 μM, 15 μM pretreatment for 12 h	antioxidation	↑ cells viability, ↓ ROS, ↑ SOD, CAT, ↑ PKA/CREB/BDNF	[56]
UroB	Neuro-2a cells	H_2_O_2_ (250 μM) for 2 h	20 μg/mL, 40 μg/mL, and 60 μg/mL pretreatment for 24 h	antioxidation	↑ cells viability, ↓ ROS, ↓ apoptosis, cytotoxicity, ↓ caspase 3, ↑ Bcl-2	[57]
UroB	BV-2 cells	LPS (100 ng/mL) or LTA (10 μg/mL) or poly(I:C) (25 μg/mL) for 16 h	30 μM, 50 μM, or 100 μM pretreatment for 1 h	antioxidation, anti- inflamation	↓ NO, ROS, TNF-α, IL-6, IL-1β, iNOS, COX-2, ↓ NF-κB, p-JNK, p-ERK, p-Akt, AP-1, ↑ IL-10, pAMPK, p47^phox^, gp91^phox^	[58]
UroA mUroA UroB mUrOA	BV-2 cells, SH-SY5Y cells	H_2_O_2_ (100 μM) for 6 h; LPS (1 μg/mL) for 24 h	0.1 μM, 0.5 μM, 5 μM, 10 μM pretreatment for 1 h, 24 h, or 48 h	antioxidation, anti- inflammation	↓ apoptosis, ↓ NO, TNF-α,NO, COX-2, IL-1, IL-6, PGE2, ↓ caspase 3/7 and 9, ↓ oxidative stress	[19]
UroA	BV-2 cells	LPS (500 ng/mL) for 3 h, 12 h, 24 h; IL-4 (100 ng/mL), IL-13 (10 ng/mL) for 24 h	10 μM pretreatment for 12 h	anti- inflammation	↓ IL-6, IL-1β, TNF-α, ↓ NOS, ↓ JNK/c-Jun, ↑ M2 microglia polarization	[59]
UroA	BV-2 cells	LPS (1 μg/mL) for 6 h, 12 h or 24 h	2.5 μM, 5 μM, and 10μM pretreatment for 2 h	anti- inflammation improved mitochondrial function	↓ IL-1β, iNOS, COX-2, ↓ ROS, ↑ MMP, ↑ p62, ↓ LC3-II, ↑ Parkin, PINK, ↓ caspase 1, NLRP3, ↓ TOM20, Tim23,↑ mitophagy, ↑ OXPHOS	[14]
UroA, UroB, UroC, mUroA dmUroC	BV-2 cells	LPS (100 ng/mL) for 30 min, 16 h or 24 h	3 μM, 10 μM, 30 μM treatment for 30 min, 1 h, 16 h or 24 h	anti- inflammation	↓ NO, TNF-α, IL-6, IL-1β, iNOS, COX-2, ↓ pAkt,↓ pERK1/2, p38 MAPK, ↓ NF-κB	[60]
UroA	BV-2 cells, ReNcell VM cells	LPS (100 ng/mL) for 1 h, 24 h, or transfected with APPSwe	2 μM, 5 μM, 10 μM treatment for 30 min, 6 h or 48 h	anti- inflammation, anti-Aβ	↑ cells viability, ↓ NO, TNFα, IL-6, ↓ Aβ, ↑ SIRT1, ↓ NF-Κb, ↑ induction of autophagic flux	[21]
UroA	SH-SY5Y cells, iPSC-ND cells	D-glucose 25 mM for 24 h, 48 h, and 72 h, Aβ _(1–42)_ for 24 h, 48 h, and 72 h	100 nM pretreatment for 30 min	anti-Aβ, improved mitochondrial function	↓ APP, BACE1, TGM2, Aβ_(1–42)_, mitochondrial calcium influx, AhR, mtROS, ↓ LDH release	[61]
UroA	SH-SY5Y cells	transfected with the APP 695	1 µM,10 µM treatment for 1 h or 24 h	improved mitochondrial function	↑ MMP, ATP, ROS, OXPHOS, mitochondrial biogenesis	[62]
UroA	PC12 cells	6-OHDA (150 μM) for 18 h or 24 h	2.5 μM, 5 μM, 10 mM treatment for 2 h	improved mitochondrial function	↑ cells viability, MPP PGC-1α, SIRT1, TFAM, ↓ apoptosis, ↑ APP, ↓ ROS, ↓ mitochondria damage	[63]
UroA DHA+ LUT+ UroA	BE(2)-M17 cells	oligomeric Aβ_1–42_ (20 μM) for 72 h	5 μM to 40 μM pretreatment for 24 h, 5 μM (combination) pretreatment for 24 h	anti-Aβ	↑ cells viability, ↓ LDH release	[64]
UroA, UroA+ EGCG	HT22 cells	transfected with APP cDNA for 24 h	no concentration mentioned, treatment for 24 h	improved mitochondrial function	↑ mitochondrial respiration	[65]
UroA, UroA+ EGCG	HT22 cells	transfected with Tau cDNA for 24 h	1 μM or 10 mM treatment for 24 h	improved mitochondrial function	↓ Drp1 and Fis1, ↑ PGC-1α, Nrf1, Nrf2, ↑ TFAM, PINK1, Parkin, ↑ Mfn1, Mfn2, and Opa1	[66]
UroA	HT22 cells	transfected with APP cDNA for 24 h	1 μM, 2 μM, 5 μM, 10 mM treatment for 24 h	improved mitochondrial function	↓ Drp1 and Fis1, ↑ PGC-1α, Nrf1, Nrf2, ↑ TFAM, PINK1, Parkin, ↑ Mfn1, Mfn2, and Opa1	[67]

Abbreviations: dmUroC: 8,9-dimethyl-O-Urolithin C, MAO: monoamine oxidase, DPPH: 1,1-diphenyl-2-picrylhydrazyl, ROS: reactive oxygen species, CAT: catalase, GR: glutathione reductase, GSH-Px: glutathione peroxidase, LDH: Lactic dehydrogenase, Bcl-2: B-cell lymphoma-2, Bax: Bcl-2-associated X, TNF-α: tumor necrosis factor-α, iNOS: inducible nitric oxide synthase, IL-6: interleukin-6, IL-1β: interleukin-1β, IL-1: interleukin-1, COX-2: cyclooxygenase-2, PARP: poly ADP-ribose polymerase, MAPK: mitogen-activated protein kinase, REDOX: mitochondrial oxidation-reduction, PKA: protein kinase A, CREB: cAMP-response element binding protein, BDNF: brain derived neurotrophic factor, NLRP3: negative regulation of NLR family pyrin domain containing 3, PGE2: prostaglandin E2, JNK: c-Jun N-terminal kinase, LC3-II: protein light chain 3-II, PINK1: PTEN induced kinase 1, MMP: mitochondrial membrane potential, OXPHOS: oxidative phosphorylation, APP: amyloid precursor protein, TGM2: transglutaminase type 2, PGC-1α: peroxisome proliferator-activated receptor-gamma coactivator-1-alpha, AhR: aryl hydrocarbon receptor, TFAM: transcription factor A, ERK1/2: extracellular signal-regulated kinase 1/2, SIRT1: silent information regulator of transcription 1, BACE1: β-site APP cleaving enzyme-1, poly(I:C): polyinosinic-polycytidylic acid, Drp1: dynamin-related protein 1, Fis1: fission mitochondrial 1; Nrf1: nuclear respiratory factor 1, Nrf2: nuclear respiratory factor 2, Mfn1: mitofusin, Mfn2: mitofusin 2, Opa1: optic atrophy 1.

**Table 2 nutrients-15-03884-t002:** Preclinical studies of Uros on brain aging in vivo animal models.

Uros	Animal	Pharmacological or Genetic Interventions	Route of Administration	Treatment (Dosage and Time)	Effects	Findings	Refs.
UroA	male ICR mice (4–6 weeks, 18–22 g)	D-gal 150 mg/kg/d s.c. for 8 weeks	i.g.	50, 100, 150 mg/kg b.w./day for 8 weeks	anti-brain aging, anti- inflammation, antioxidation	↑ spontaneous locomotion, object recognition learning, ↓ AchE, MAO, ↑ SOD, CAT, GSH-Px, ↓ p53/p21, TEAC, ↑ SIRT1, ↓ TNF-α, IL-1β, and IL-6, ↑ Bcl-2, ↓ caspase 3, mTOR, ↓ dysfunctional autophagy, astrocyte activation, ↓ apoptosis	[17]
UroB	male C57BL/6 mice (6–8 weeks, 18–22 g)	D-gal 150 mg/kg/d s.c. for 8 weeks	i.g.	50, 100, 150 mg/kg b.w./day for 8 weeks	anti-brain aging, anti- inflammation, antioxidation	↓ cognitive deficits, ↑ pAkt, ↑ hippocampal LTP,↑ CAT, GSH-Px, TEAC, SOD, ↓ MDA, ↓ TNF-α, IL-6, IL-1β, AGEs, cytotoxicity, ↓ the activation of microglia and astrocytes, ↓ AchE, ↑ number of neuron, ↓ MAO	[57]
UroA	female APP/PS1 transgenic mice (28 weeks)	transgenic AD mice	i.g.	300 mg/kg b.w./day for 14 days	anti- inflammation	↓ spatial learning deficits, ↑ neurogenesis, ↓ neuronal apoptosis, ↓ reactive gliosis, ↓ Aβ, IL-1β, TNF-α, ↑ AMPK, ↓ p-P65, NF-κB, p-P38, MAPK, BACE1	[22]
UroA	APP/PS1 transgenic mice (13 months) *C.elegans*	transgenic AD mice	i.g.	200 mg/kg b.w./day for 1 month or 0.1 mM (*C.elegans*)	anti- inflammation	↑ learning and memory retention, ↑ OCR, ↓ ROS, ↓ Aβ_1–42_, Aβ_1–40_, ↑ IL-10, ↓ autophagy, ↓ IL-6, TNF-α, ↓ NLRP3, IL-1β, ↓ p-tau, caspase 1	[68]
UroA	CX3CR1- Cre mice	MPTP 15 mg/kg/d, i.p. 4 times a day every2 h	i.g.	20 mg/kg b.w./day for 7 days	anti- inflammation	↓ motor deficits, ↑ TH, ↓ caspase 1, NLRP3, ↓ astrogliosis	[14]
UroB	Male ICR mice (7 weeks, 2–37 g)	LPS 5 mg/kg/d, i.p.	i.p.	50 mg/kg b.w./day for4 days	anti- inflammation	↓ microglia activation, ↓ NADPH, Akt, JNK, ERK, ↑ AMPK, HO-1	[58]
UroA	male C57BL/6J mice (8–10 weeks)	6-OHDA 9 µg	i.p.	10 mg/kg b.w./day for 7 days	improved mitochondria function	↓ neurotoxicity, mitochondria damage, OXPHOS, ↑ PGC-1α, TFAM, ↑ SIRT1	[63]
UroA	mice	STZ 75 mg/kg/d i.p. for 3 days	i.p.	2.5 mg/kg b.w./day for 8 weeks	anti-Aβ, improved mitochondria function	↓ APP, BACE1, p-tau, Aβ_(1–42)_, TGM2	[61]
UroA UroB mUroA mUroB	*C. elegans*	transgenic AD *C. elegans* (CL4176)	feeding	10 μg/mL pretreatment for 20 h	anti-Aβ	↑ *C. elegans* survival and mobility	[69]
UroA UroA + EGCG	hAbKI mice (3 months)	humanized homozygous Aβ knockin (hAbKI) AD mice	i.p.	UroA 2.5 mg/kg b.w., EGCG 25 mg/kg b.w., 3 times per week for 4 months	anti-Aβ, improved mitochondria function	↑ mitochondrial fusion, synaptic, ↓ Aβ_(1–40)_ and Aβ_(1–42)_, ↑ mitophagy, autophagy genes, ↓ mitochondrial fission genes, mitochondrial dysfunction, ↑ dendritic spines, ↓ fragmented mitochondria number, ↑ mitochondrial length, mitophagosomal formations	[65]
UroA	male C57BL/6J mice (5 weeks, 18–22 g)	STZ 30 mg/kg b.w./day i.p. for 4 days	i.g.	200 mg/kg b.w.	anti-brain aging, anti- inflammation	↓ hyperglycemia, ↑ learning and memory, ↓ IL-6, IL-1β, TNF-α, IL-1β, COX-2, iNOS-2, ↑ IL-10, ↓ NLRP3	[70]

Abbreviations: i.p.: intraperitoneal, s.c.: subcuta, i.g.: intragastrical, b.w.: body weight, AchE: acetylcholinesterase, MAO: monoamine oxidase, SOD: superoxide dismutase, CAT: catalase, GSH-Px: glutathione peroxidase, AGEs: advanced glycation end products, NADPH: triphosphopyridine nucleotide, TEAC: Trolox-equivalent antioxidant capacity, SIRT1: silent information regulator of transcription 1, TNF-α: tumor necrosis factor-α, iNOS: inducible nitric oxide synthase, IL-6: interleukin-6, IL-1β: interleukin-1β, IL-1: interleukin-1, COX-2: cyclooxygenase-2, Bcl-2: B-cell lymphoma-2, LTP: long-term potentiation, NLRP3: negative regulation of NLR family pyrin domain containing 3, MDA: malondialdehyde, BACE1: β-site APP cleaving enzyme-1, OCR: oxygen consumption rate, TH: tyrosine hydroxylase, HO-1: heme oxygenase 1, TFAM: transcription factor A, STZ: streptozotocin, TGM2: transglutaminase type 2.

**Table 3 nutrients-15-03884-t003:** Clinical studies of Uro precursors on brain aging.

Source	Subjects	Clinical Trial Procedure	Treatment (Dosage and Time)	Effects	Fundings	Refs
Pomegranate juice	Age: 54–72 years; Cognition and memory: age-related memory decline; Other heath state: no neurological, psychiatric and major medical conditions	Randomized, placebo controlled, double blind trial	Dosage: 240 mL/day of pomegranate juice (*n* = 15) or placebo drink (*n* = 13) Time: 4 weeks	anti-age-related memory decline	↑ fMRI activity during verbal and visual memory tasks, ↑ memory ability, ↑ plasma antioxidant status	[42]
Pomegranate juice	Age: 50–75 years; Cognition and memory: age-related memory decline; other heath state: no cerebrovascular disease, neurological or physical illnesses associated with cognitive deterioration	Randomized, placebo controlled, double blind trial	Dosage: 236.5 mL/day of pomegranate juice (*n* = 98) or placebo drink (*n* = 102) Time: 48 weeks	anti-age-related memory decline	↑ visual memory, ↑ visual learning and recall, ↑ verbal memory, words recall	[49]
Nuts	Age: 55–80 years; Cognition and memory: healthy; other heath state: no diabetes, smoking, hypertension, dyslipidemia, overweight and cardiovascular disease	Randomized, placebo controlled trial	Dosage: MedDiet + EVOO 1 L/week (*n* = 224); MedDiet + nuts 30 g/day (*n* = 166); or low-fat diet (*n* = 132) Time: 6.5 years	anti-age-related memory decline	↑ orientation to time and place, ↑ registration, attention and calculation, ↑ recall, language, and visual construction, ↑ visuospatial abilities, working memory, attention, ↑ abstract thinking, language comprehension	[71]
Walnuts	Age: 63–79 years; Cognition and memory: healthy; other heath state: no neurodegenerative disease, stroke, head trauma, brain surgery, psychiatric illness, depression, obesity, diabetes, hypertension and chemotherapy	Randomized controlled trial	Dosage: Walnuts 30–60 g/day (*n* = 336) or control diet (abstention from walnuts) (*n* = 321) Time: 2 years	anti-age-related memory decline	↑ global cognition and perception	[51]
Strawberry	Age: 60–75 years; Cognition and memory: age-related motor and cognitive decline; other heath state: BMI (18.5–29.9), no psychological or psychiatric disorders and chronic disease	Randomized, placebo controlled, double blind trial	Dosage: Strawberry 24 g/day (*n* = 18) or placebo (*n* = 19) Time: 45 or 90 days	anti-age-related memory decline	↑ words recalled, verbal learning	[72]
Mixture of berries	Age: 50–70 years; Cognition and memory: healthy; other heath state: no metabolic disorders, food allergies and, gastrointestinal disorder	Randomized cross-over trial	Dosage: mixture of berries (150 g blueberries, 50 g blackcurrant, 50 g elderberry, 50 g lingonberries, 50 g strawberry, and 100 g tomatoes/day) (*n* = 20); or placebo drink (*n* = 21) Time: 5 weeks	anti-age-related memory decline	↑ verbal working memory, ↑ selective attention, ↓ total- and LDL cholesterol, ↑ insulin concentrations	[73]
Grape and blueberry extract	Age: 60–70 years; Cognition and memory: age-related memory decline; other heath state: BMI (20–30)	Randomized, placebo controlled, double blind trial	Dosage: grape and blueberry extract 600 mg/day (*n* = 91) or placebo (*n* = 98) Time: 6 months	anti-age-related memory decline	↑ verbal episodic, ↑ recognition memory, ↑ working memory	[74]
Blueberry and blueberry extract	Age: 65–80 years; Cognition and memory: age-related memory decline; other heath state: no metabolic disorders and diabetes	Randomized, placebo controlled, double blind trial	Dosage: blueberry 500 mg/day (*n* = 28); blueberry 1000 mg/day (*n* = 29); blueberry extract 111 mg/day (*n* = 28); or placebo (*n* = 27) Time: 6 months	anti-age-related memory decline	↑ word recognition, ↑ total number of sequences correctly recalled, ↓ systolic blood pressure	[75]
Blueberry	Age: >65 years; Cognition and memory: age-related memory decline other heath state: healthy	Pilot, single-blind, one-arm trial	Dosage: blueberry 444 mL/day (weighing 54–64 kg); 532 mL/day (weighing 54–64 kg); 621 mL/day (weighing 77–91 kg); (*n* = 9) Time: 12 weeks	anti-age-related memory decline; antidepressant	↑ paired associate learning, ↑ word list recall, ↓ depressive symptoms, ↓ fasting glucose levels	[50]
Blueberry	Age: <65 years; Cognition and memory: healthy other heath state: no contraindications to fMRI	Randomized, placebo controlled, double blind trial	Dosage: blueberry 30 mL/day (*n* = 12) or placebo drink (*n* = 14) Time: 12 weeks	anti-age-related memory decline	↑ brain perfusion and activation, ↑ psychomotor function, visual processing, executive function, verbal and spatial memory	[52]
Blueberry	Age: 62–80 years; Cognition and memory: age-related memory decline; other heath state: no diabetes, kidney disease, liver disease, hematological coagulation disorder	Randomized, parallel groups, placebo controlled, double blind trial	Dosage: fish oil (1.6 g EPA + 0.8 g DHA/day) (*n* = 15); blueberry 25 g/day (*n* = 16); fish oil + blueberry 24 g/day (*n* = 17); or placebo (*n* = 17) Time: 24 weeks	anti-age-related memory decline	↑ psychomotor speed, working memory, ↑ lexical access, ↑ long-term memory	[53]
Frozen blueberry	Age: 60–75 years; Cognition and memory: healthy; other heath state: BMI (18.5–29.9), no smoking or use of medications	Randomized, placebo controlled, double blind trial	Dosage: frozen blueberry 24 g/day (*n* = 19) or placebo (*n* = 19) Time: 90 days	anti-age-related memory decline	↑ executive function, ↑ long-term memory, short term memory, ↑ spatial cognition, and attention	[76]
Frozen blueberry	Age: 68–92 years; Cognition and memory: age-related memory decline; other heath state: no serious psychiatric disorder, substance abuse, and claustrophobia	Randomized, placebo controlled, double blind trial	Dosage: frozen blueberry 25 g/day (*n* = 8) or placebo (*n* = 8) Time: 16 weeks	anti-age-related memory decline	↑ working memory, accuracy, ↑ blood oxygen level dependent activation	[77]
Frozen blueberry	Age: >68 years; Cognition and memory: age- related memory decline; other heath state: no dementia, serious psychiatric condition, substance abuse	Randomized, placebo controlled, double blind trial	Dosage: frozen blueberry 24 g/day (*n* = 16) or placebo 20 g/day (*n* =21) Time: 16 weeks	anti-age-related memory decline	↑ lexical access for semantic information, ↑ speed of processing and working memory, ↑ verbal and nonverbal long-term memory	[78]

Abbreviations: fMRI: functional magnetic resonance imaging, BMI: body mass index, EVOO: extra-virgin olive oil, MedDiet: Mediterranean diet, LDL: low-density lipoprotein, EPA: eicosapentaenoic acid, DHA: docosahexaenoic acid.

## 4. Mechanisms of Action

The beneficial effects of Uros on brain function during aging are associated with multi-target actions that involve relieving chronic oxidative stress and inflammation, promoting mitophagy and mitochondrial function, inhibiting amyloid-β (Aβ) and tau pathology, and regulating tryptophan (Trp) metabolism. The mechanisms are summarized in Figure 1.

### 4.1. Antioxidant Activity in CNS

As the brain consumes more than one-fifth of the total oxygen, the oxidative stress exerted by reactive oxygen species (ROS) and related degeneration is particularly severe in the brain owing to aging. Damage to neural cells occurs when ROS production overwhelms the antioxidant defense mechanisms. Endogenous antioxidant defenses in the brain are relatively low compared to those in other vital organs [79,80]. This makes the protection of antioxidants, which can pass through the BBB, particularly important for the progression of brain aging. Although ETs and EA cannot cross the BBB, Uros are potent antioxidants with good BBB permeability [36,69].

First, UroA and UroB are evidenced as direct radical scavengers (details listed in Table 4) [54,57]. Furthermore, cell models have shown that UroA and UroB protected neuron-like cells from direct H_2_O_2_-induced damage, including PC12 cells, SK-N-MC cells, SH-SY5Y cells, and Neuro-2a cells, in which they effectively inhibited ROS formation and lipid peroxidation [17,20,54,55,57,81]. Moreover, Uros can boost endogenous antioxidant defenses in neuronal cells and brains of aged mice. Data from an array of studies have shown that UroA and UroB increased the activity of antioxidant enzymes, including catalase, superoxide dismutase, glutathione reductase, and glutathione peroxidase [58,82]. UroA increases the expression of cytoprotective peroxiredoxins 1 and 3 in Neuro-2a cells [54]. A study on UroB demonstrated that pretreatment with UroB significantly decreased the mRNA expression of nicotinamide adenine dinucleotide phosphate oxidase subunits (p47^phox^ and gp91^phox^) and increased the antioxidant hemeoxygenase-1 expression via nuclear factor erythroid-2 related factor 2/antioxidant response element signaling in lipopolysaccharide (LPS)-treated BV2 cells [58].

Additionally, Uros are inhibitors of several oxidases (pro-oxidant enzymes) that can promote ROS formation. It is shown that Uros inhibited the activity of monoamine oxidase (MAO), which is responsible for the metabolism of monoamine neurotransmitters such as serotonin and dopamine [54,57,83]. A recent study demonstrated that Uros (A, B, and C) were selective inhibitors of human MAO-A (rather than MAO-B), and UroB was the strongest inhibitor with the IC_50_ of 0.88 ± 0.24 μM (the IC_50_ of UroA and UroC was 5.88 ± 0.69 μM and 29.6 ± 1.8 μM, respectively), whereas EA had no effect on MAO-A [83]. Clinically, MAO inhibitors may alleviate the symptoms of depression and Parkinson’s disease (PD). Therefore, these findings may suggest the potential benefits of Uros for related disorders. Saha et al. reported that UroA significantly inhibited heme peroxidase activity, including myeloperoxidase and lactoperoxidase in in vitro and in vivo models [84], which may provide a better understanding of the peroxidase inhibitory and anti-inflammatory activities of UroA. 

The antioxidant effects of Uros are beyond the traditional antioxidant activities of their precursors. ETs and EA are typical antioxidants, which are mainly attributed to their potent free radical scavenging activity, including a wide variety of ROS and reactive nitrogen species [85,86]. As the bioavailability of ETs and EA is poor, the high antioxidant capacity of dietary ETs and EA may be important and restricted mostly to local actions in the GI tract [25,87]. González-Sarrías et al. suggested a lower neuroprotective activity of Uros against oxidative stress-induced cell death than that of their precursors [20,59]. It appears that Uros are not as potent antioxidants as their precursors, at least in vitro, and may systemically exert their antioxidant activity. Recently, we found that UroA (5, 10 μM) treatment significantly increased protein kinase A (PKA)/cAMP-response element binding protein (CREB)/brain derived neurotrophic factor (BDNF) neurotrophic signaling pathway in H_2_O_2_-treated SH-SY5Y cells, and pretreatment with PKA inhibitor H89 abolished the protective effects of UroA in H_2_O_2_-treated SH-SY5Y cells [56]. These results indicated that PKA/CREB/BDNF neurotrophic signaling pathway might involve the neuroprotective effect of UroA against oxidative stress.

### 4.2. Mitigation of Neuroinflammatioin

Chronic inflammation caused by innate immune cells such as glia is thought to play a key role in brain aging and related diseases. Glial activation and the release of inflammatory molecules such as proinflammatory cytokines are hallmarks of chronic inflammation in the brain. Mitigating neuroinflammation is one of the most important mechanisms underlying the health benefits of Uros.

It has been well documented that UroA and UroB significantly attenuated the inflammation induced by LPS in mouse microglia BV2 cells, including inhibiting the production of NO and the expression of inducible nitric oxide synthase (iNOS) and cyclo-oxygen-ase-2 (COX-2), regulating the levels of proinflammatory cytokines such as interleukin (IL)-6, IL-1β, tumor necrosis factor-α (TNF-α), and anti-inflammatory cytokines such as IL-10 [19,21,58,60,88]. In addition to LPS, Lee et al. reported that UroB significantly inhibited lipoteichoic acid (LTA)- and polyinosinic-polycytidylic acid (poly(I:C))-induced inflammation in BV2 cells, which are known as toll-like receptor (TLR) 2 and TLR3 agonists, respectively [58]. These results suggested that the anti-inflammatory effect of UroB is not confined to LPS stimulation. Moreover, UroA can induce M2 polarization from the M1 state in microglia and other macrophage-like cells which can subsequently inhibit pro-inflammatory cytokines generation and promote neuroprotection [59,89]. Animal studies have shown that the administration of UroA and UroB could significantly improve behavioral deficits and neuroinflammation in D-galactose-induced aged mice, transgenic APP/PS1 mice, LPS-brain-injected mice, and 1-methyl-4-pheny1-1,2,3,6-tetrahydropyridine (MPTP)-induced PD mice, including decreasing the level of pro-inflammatory cytokines in the brain and inhibiting the activation of microglia and astrocytes in the hippocampus and cortex [14,17,22,57,58]. In addition, UroA was evidenced to reduce the expression and activity of the NACHT, LRR, and PYD domains-containing protein 3 (NLRP3) inflammasome in LPS-treated BV2 cells and mouse models (APP/PS1 mice and MPTP-treated mice) [30,68].

Multiple proteins and signaling pathways such as adenosine monophosphate (AMP)-dependent kinase (AMPK), mitogen-activated protein kinase (MAPK), nuclear factor-κB (NF-κB), and silent information regulator of transcription 1 (SIRT1) are considered to participate in the anti-inflammatory effects of Uros [58,68,88,89,90,91,92,93,94]. The NF-κB pathway is the best-studied inflammatory signaling pathway and is thought to be critical for the synthesis and effects of pro-inflammatory cytokines. An experiment in LPS-induced BV2 cells showed that UroA and UroB significantly inhibited NF-κB p65 expression and nuclear translocation in a dose-dependent manner and that UroB displayed more potent inhibitory activity than UroA [60]. AMPK, a central regulator of energy and metabolism involved in the pathophysiology of aging and age-related diseases, plays an important role in chronic inflammation. AMPK activation exerts anti-inflammatory effects by regulating cytokine synthesis and multiple inflammatory signaling pathways, such as NF-κB and MAPK pathways [95,96,97]. Moreover, AMPK signaling regulates NLRP3 inflammasome formation and activation during aging [98]. In addition to its established role in inflammation, AMPK activation has recently been implicated in promoting microglial M2 polarization, thereby relieving brain inflammation [99]. Experiments in cell and animal models have indicated that AMPK activation is tightly involved in the anti-inflammatory effects of Uros. In LPS-treated BV2 cells, UroB increases AMPK phosphorylation and decreases the phosphorylation of its downstream molecules Akt, JNK, and ERK [58,60]. In APP/PS1 mice, Zhuo et al. found that UroA treatment significantly upregulated AMPK signaling and downregulated NF-κB and MAPK signaling in the cortex and hippocampus, which was responsible for the improvement of memory and neuroinflammation [22]. SIRT1, a histone deacetylase, plays a key role in regulating neuroinflammation and the release of pro-inflammatory cytokines [100,101,102,103]. SIRT1 activation inhibits NF-κB signaling by involving deacetylation of the p65 subunit [104]. A study on UroA showed that treatment with UroA (5 and 10 μM) significantly increased the level (nuclear) and the activity (cell) of SIRT1 in BV2 cells, and the anti-inflammatory effect of UroA was abolished in the presence of EX527, a SIRT1 inhibitor, suggesting that the activation of SIRT1 is required for the anti-inflammatory effect of UroA [21].

Additionally, to compare the anti-inflammatory effect of UroA and its precursors, Ashley et al. investigated the difference between whole red raspberry polyphenols (RRW) and UroA in BV-2 cells under 3 h, 12 h, and 24 h inflammatory conditions (LPS and ATP treatment). The results demonstrated that RRW only inhibited the inflammation induced by 3 h of LPS stimulation, whereas UroA inhibited the inflammation caused by LPS treatments (3, 12, and 24 h). Moreover, the anti-inflammatory effects of RRW and UroA were both mediated by downregulation of the JNK/c-Jun pathway [59].

### 4.3. Promotion of Mitophagy and Mitochondrial Function

The brain possesses high mitochondrial activity to meet its relatively high energy demands. Mitochondrial dysfunction, including the abnormalities of mitochondrial bioenergetics, mitochondrial biogenesis, and autophagy in neurons, accumulate during aging. An increasing number of studies have indicated that the maintenance of mitochondrial homeostasis is particularly important for brain function in the elderly and may be a potential therapeutic strategy for age-related neurodegenerative diseases. Accumulating evidence suggests that UroA is an important regulator of mitochondrial homeostasis, particularly a robust mitophagy inducer, which seems to be one of the crucial mechanisms underlying its health benefits against aging [61,105]. Ryu et al. were the first to report that UroA extended the lifespan of *C. elegans* and improved muscle function in mice through mitophagy activation [106]. Recently, an increasing number of clinical trials have confirmed the impact of UroA on mitophagy in humans. Results from two clinical trials in elderly individuals demonstrated that supplementation with UroA at doses of 500 mg and 1000 mg daily for 4 weeks or 4 months significantly improved mitophagy and mitochondrial biogenesis in skeletal muscles [37]. Moreover, another recent clinical trial in middle-aged adults showed that supplementation with UroA increased mitophagy proteins and reduced inflammation [38]. Fang et al. conducted a systematic study on the promotion of mitophagy by UroA and found that mitophagy induction underlies the multiple actions of UroA in the brain, including the improvement of cognition and memory deficits, the alleviation of chronic neuroinflammation, and the inhibition of Aβ and tau pathology [68,107]. In vitro study demonstrated that UroA increased the levels of a series of mitophagy-related proteins, including phosphatase and tensin homolog deleted on chromosome ten induced kinase 1 (PINK1), Parkinson’s disease-related-1, beclin-1, Bcl-2-like protein 13, activating molecule in BECN1-regulated autophagy protein 1, and serine/threonine-protein kinase ULK1 (ULK1) in human neuronal SH-SY5Y cells. Moreover, UroA inhibited the phosphorylation of tau (p-tau) in a mitophagy-dependent manner, and the inhibitory effect was diminished in PINK1 or ULK1 knockdown cells. Consistently, in vivo studies showed that UroA activated neuronal mitophagy and reduced mtROS levels and mitochondrial damage in the hippocampus and prefrontal cortex of APP/PS1 mice, and mitophagy induction was also required for memory improvement of UroA in tau transgenic *C. elegans* and mice [68]. More recently, a study in humanized homozygous Aβ knockin (hAbKI) mice of late-onset AD demonstrated that UroA administration for 4 months significantly improved phenotypic behavior changes and mitochondrial dysfunction, including mitochondrial bioenergetics, biogenesis, and mitophagy [65].

Similar to neuronal cells, UroA promotes mitophagy and improves mitochondrial dysfunction in microglial cells. Treatment with the autophagy inhibitor 3-methyladenine abolished the effect of UroA on mitochondrial dysfunction and inflammation in LPS-treated BV2 cells [14,21]. Studies in animals demonstrated that mitophagy in microglial cells decreased by 60% in the hippocampus of AD mice, whereas UroA treatment normalized the decreased mitophagy, thereby increasing the phagocytic efficiency of microglia and the removal of Aβ plaques [68]. Similarly, UroA reduced the elevated expression and activity of NLRP3 and related neuroinflammation in AD mice by inducing mitophagy in the microglia. Moreover, using Atg5flox/flox:CX3CR1-Cre mice to block microglial mitophagy in vivo, the improvement in behavioral deficits and neuroinflammation induced by UroA in PD mice was diminished [14].

However, studies on the autophagy and mitophagy of UroA have also yielded conflicting results. Research conducted by Esselun et al. revealed that UroA did not affect mitophagy in SH-SY5Y-APP695 cells but moderately promoted mitochondrial biogenesis [62]. Another recent study indicated UroA induced robust neuronal mitophagy but did not increase neuronal macroautophagy in transgenic nematodes [68]. Additionally, Ahsan et al. demonstrated that UroA activated autophagy but not mitophagy in ischemia/reperfusion injured neuronal cells and mice [108].

Studies have indicated that UroA activated autophagy and mitophagy by regulating several key signaling molecules, including AMPK, peroxisome proliferator-activated receptor-gamma coactivator-1-alpha (PGC-1α), and SIRT, and inhibition of the mammalian target of rapamycin (mTOR). These signaling pathways stimulate mitophagy and mitochondrial biogenesis and are regulated by Uros. As discussed in the anti-inflammatory section, AMPK is one of the most important targets in aging and has been well documented in age-related mitochondrial dysfunction [109,110,111]. The downregulation of AMPK activity during aging impairs metabolic regulation, increases oxidative stress, and reduces autophagic clearance [98]. Activated AMPK increases PGC-1α level, which directly increases mitochondrial biogenesis [112]. The key autophagy/mitophagy effectors, such as SIRT, mTOR, and ULK1, are all downstream molecules of AMPK signaling [113,114,115]. Moreover, SIRT can deacetylate and activate liver kinase B1 (LKB1), an upstream kinase that activates AMPK [116]. Recent studies have demonstrated that Uros activate AMPK activity in a variety of cell types in vitro, including microglia, macrophages, and nucleus pulposus cells [22,58,117], as well as in multiple tissues in vivo, including the brain, muscle, and pancreas [92,118,119], indicating that Uros may be an AMPK activator. Therefore, AMPK may be a crucial target underlying the beneficial effects of Uros on aging, which requires further investigation.

Additionally, UroA activates SIRT1 and SIRT3. UroA stimulates SIRT3 promoter activity in Caco-2 cells [120] and increases ATP and NAD+ levels, leading to activation of the SIRT1-PGC-1α pathway in murine muscle [121]. Moreover, Chen et al. reported that UroA improved impaired autophagy in D-galactose-induced mice by upregulating SIRT1 signaling and downregulating mTOR signaling, and these effects appear to be mediated by the activation of miR-34a [17]. Treatment with both SIRT1 inhibitor EX527 and autophagy inhibitor chloroquine abolished the neuroprotective effect of UroA in APPSwe-transfected ReNcell VM human neural cells, as well as the anti-inflammatory effect of UroA in LPS-treated BV2 cells [21]. SIRT1 increases mitofusin 2 (Mfn2) and Parkin levels during mitophagy activation [122,123], and UroA has been shown to increase Mfn2 and Parkin expression in humans after 4 weeks of administration [37]. In addition, UroA activates autophagy by inhibiting endoplasmic reticulum (ER) stress in cell and mouse models of ischemic neuronal injury [62]. In summary, mitophagy deficiency and subsequent mitochondrial dysfunctions are early and pivotal events in brain aging and neurodegenerative diseases, and mitophagy or autophagy induction seems to be a crucial mechanism in the effects of Uros.

### 4.4. Inhibition of Aβ and Tau Pathology

Abnormal accumulation of Aβ and neurofibrillary tangles of p-tau protein are hallmark features of AD. It has been well documented that Aβ deposition in the brain occurs approximately two decades before the onset of the disease. Several studies have shown that Uros inhibited Aβ deposition and p-tau generation during brain aging [22,61,68,69]. The thioflavin T (ThT) binding assay in vitro showed that UroA, UroB, methyl-UroA (mUroA), and mUroB inhibited Aβ fibrillation including Aβ fibril content, β-sheet formation, and peptide oligomerization, whereas only mUroB reversed the neurotoxicity and muscular paralysis in the Aβ_1–42_-induced *C. elegans* model of AD [69]. A more recent study in vitro demonstrated that UroA inhibited Aβ_1–42_-induced toxicity in human neuroblastoma BE(2)-M17 cells [64]. In transgenic APP/PS1 AD mice, immunohistochemistry and ELIAS results demonstrated that UroA treatment significantly decreased Aβ_1–40_ and Aβ_1–42_ plaques in the cortex and hippocampus [22]. Moreover, UroA administration for 4 months significantly reduced the levels of Aβ_1–40_ and Aβ_1–42_, and improved inflammation, synaptic structure, and function in the brain of hAbKI AD mice [65]. In a further study conducted by Fang et al., UroA treatment restored memory impairment in both Aβ nematodes expressing pan-neuronal human Aβ_1–42_ (CL2355) protein and tau nematodes expressing pan-neuronal tau fragments (BR5270) [68]. Aβ plaque formation is a result of imbalance between Aβ clearance and its production from amyloid precursor protein (APP), and cleavage of APP by β-site APP cleaving enzyme-1 (BACE-1) is responsible for Aβ production. Although the precursors of Uros, such as EA and punicalagin, have been reported to inhibit the activity of BACE1 [124,125], UroA was shown to decrease only the expression of BACE1 in mice, and the effects of UroA on APP expression are inconsistent [61,62]. Additionally, microglia play a crucial role in the clearance of extracellular Aβ plaques [126,127]. In a study by Fang, UroA administration enhanced the removal of insoluble Aβ_1–42_ and Aβ_1–40_ proteins in the hippocampus of APP/PS1 mice by increasing the phagocytic efficiency of microglia, whereas no significant effect on the levels of APP cleavage intermediates was observed, suggesting that APP proteolysis may not be the target of UroA. Furthermore, the study also revealed that UroA treatment inhibited the phosphorylation of tau at Thr181, Ser202/Thr205, and Ser262 in tau-overexpressing human SH-SY5Y cells. Consistently, UroA administration inhibited p-tau and improved memory impairment in the 3×Tg AD mice [68]. More recently, Tu et al. showed that dual-specific tyrosine phosphorylation-regulated kinase 1A (DYRK1A) is the main target of UroA in its anti-AD effect. UroA significantly inhibited the activity of DYRK1A, which led to the de-phosphorylation of tau and further stabilized microtubule polymerization [128].

Hyperglycemia and diabetes are risk factors for neurodegeneration and critical contributors to amyloidogenesis. Uros were shown to improve diabetes-associated neurodegeneration. Lee et al. systematically studied and reported that UroA attenuated the amyloidogenesis and neurodegeneration in a diabetes model. Their experiments in vitro showed that pretreatment with UroA significantly decreased high glucose-induced Aβ_1–42_ formation, as well as APP and BACE1 expression in both SH-SY5Y cells and human iPSC-derived neuronal differentiated cells. Furthermore, the experiments in vivo showed that UroA injection at dose of 2.5 mg/kg for 8 weeks improved cognitive impairment in streptozotocin-induced diabetic mice and decreased the levels of Aβ_1–42_, APP, BACE1, and p-tau (Ser262 and Ser396) in the prefrontal cortex and hippocampus [61]. In addition, Chen et al. reported that UroB supplementation improved learning and memory impairment by inhibiting the accumulation of advanced glycation end products in the brain of aged mice [57]. Xiao et al. found that UroA relieved diabetes-associated cognitive impairment and neuroinflammation by alleviating intestinal barrier dysfunction via N-glycan biosynthesis pathway [70].

### 4.5. Regulation of Trp Metabolism

Recent studies have indicated that the regulation of Trp metabolism may be a novel mechanism underlying the health benefits of UroA against brain aging. Trp metabolism is an important communication strategy in the “microbiome–gut–brain” axis, in which both host and gut microbiota are involved. Trp metabolites can serve as neurotransmitters and signaling molecules in CNS. Although it is well known that Trp is used for the synthesis of serotonin, a large majority of Trp (>95%) is metabolized by the kynurenine (Kyn) pathway, thus producing bioactive metabolites with distinct activities in CNS. In humans, the activity of Kyn pathway increases with age [129]. Several lines of evidence suggest that the imbalance in Trp metabolism is associated with various neurodegenerative and neurological diseases [130,131]. Trp metabolites can control the pathogenic activities of microglia and astrocytes via aryl hydrocarbon receptors (AhRs), and further inhibit neuroinflammation and neurodegeneration [132,133]. A recent study showed that oral administration of EA and UroA for 8 weeks significantly regulated the microbial composition and Trp metabolism in DBA/2J mice fed with a high-fat and high-sucrose diet. Both UroA and EA supplementation reduced the Kyn pathway, and UroA significantly decreased the level of indole sulfate in serum [134]. Additionally, UroA elevated Trp hydroxylase-2 transcription, the rate-limiting enzyme in Trp metabolism to serotonin, and subsequently increased serotonin production in differentiated rat serotonergic raphe cells [135]. Therefore, the regulation of UroA on Trp metabolism may be an important mechanism underlying its beneficial effect, but this has not been clarified and deserves further research.

### 4.6. Others

Uros have been evidenced to exert neuroprotection by direct action on estrogen receptors (ERs) [136,137] and AhRs [138] in the brain. They are considered as “enterophytoestrogens”, which are microflora-derived metabolites with estrogenic and/or antiestrogenic activities [139,140]. Furthermore, UroA has demonstrated high selectivity on ERα and has recently been reported to regulate ERα-dependent gene expression in endometrial cancer cells [141]. AhRs are ligand-activated transcription factors involved in multiple physiological and pathological processes. In addition to indirectly regulating AhRs through Trp metabolites, UroA has been evidenced as a human-selective AhR antagonist [142,143], which is considered to mediate its anti-inflammatory activity. Additionally, UroA can alleviate BBB dysfunction [144]. 

Although these studies have clarified the effects of Uros from different aspects, they are relatively isolated from each other. The key targets and intracellular signaling pathways underlying the effects of Uros on aging brain remain to be elucidated. Recently, pharmaceutical studies have been conducted on Uros. Several natural and synthetic Uro analogs exhibit inhibitory effects on various enzymes, including cholinesterases, MAO-B, and cyclooxygenases [145,146]. Therefore, Uros may be multi-target agents against neurodegenerative and neurological diseases.

## 5. Knowledge Gaps

First, clinical evidence for Uros or UroA on age-related cognitive decline or neurodegenerative diseases is still unclear. UroA has been designated as GRAS by FDA in 2018 and sold as an ingredient of anti-aging products in USA. Growing clinical evidence has demonstrated that precursors of Uros including pomegranate, blueberry, and walnut supplementation significantly improved age-related brain dysfunctions (Table 3). The specific advantages of UroA in preventing brain aging is mainly based on (i) good safety and tolerance, (ii) favorable blood–brain barrier permeability and selective brain distribution, (iii) the accumulating evidence from preclinical studies, (iv) the different capacities for producing Uros between individuals and the decreased capacities in the elderly, and (v) overcoming limitations and deficiencies of microbiome-based treatment. Therefore, clinical trials of UroA supplementation on cognitive and memory decline in the elderly should be carried out as soon as possible, which seem to be the crucial obstacle on the way to its application. 

Second, the pharmacokinetic features of UroA (pharmacokinetic research of Uros is mainly focused on UroA) in the brain have not been clarified. UroA is easily absorbed after oral administration and mainly metabolized to UroA glucuronide, UroA sulfate, and mUroA. Studies in rodents have shown that UroA and mUroA are detectable in the brain after administration of UroA or pomegranate juice [147]. The Uro metabolites identified in the brain are summarized in Table 5. A recent study depicted a more detailed profile of UroA in the brain, in which the concentration of UroA increased slowly after 0–3 h of oral administration (UroA, 200 mg/kg) and reached the maximum concentration (Cmax) at approximately 28 ng/g and 35 ng/g in the cortex and hippocampus, respectively, after 4 h (Tmax) of oral administration. In contrast, the Cmax and Tmax of UroA in plasma were 15 ng/mL and 2 h, respectively [148]. This study is the first to report that UroA was detectable in the specific brain regions (cortex and hippocampus) and remained at high levels for hours after a single oral administration. However, more detailed pharmacokinetic parameters should be determined and investigated.

Third, the therapeutic targets and molecular mechanisms underlying the effects of UroA against brain aging remain to be elucidated. Although UroA has been found to activate anti-aging signaling pathways such as AMPK, SIRTs, and mTOR in vivo, the direct actions of UroA on these targets are currently unclear. 

Fourth, the biotransformation of Uros in vitro is essential for their further application. Although there are anti-aging products containing high purity UroA of chemical synthesis on sale, biotransformation may greatly improve producing efficiency. This process is a two-step bioconversion. In the first step, the conversion of ETs to EA has been clarified and achieved in vitro [150]. Elucidating the second step and achieving the transformation of EA into Uros in vitro could yield promising application.

Fifth, it has been reported that UroA can attenuate diabetes-associated cognitive impairment by ameliorating intestinal barrier dysfunction [70]. However, the impact of UroA supplementation on gut microbiota ecology is unclear. The role of UroA supplementation on the gut microbiota composition and other neuroactive metabolites, e.g., SCFAs need further investigation. 

Last, recent studies in animals and cells indicated that the combination of UroA with other food functional factors such as docosahexaenoic acid (DHA) and egpigallocatechin gallate (EGCG) produced significantly synergistic effects against brain aging and AD [64,65]. The combined treatments may have referential value for UroA to develop into related functional food and drugs. However, the synergistic efficacy and mechanisms warrant further elaboration.

## Figures and Tables

**Figure 1 nutrients-15-03884-f001:**
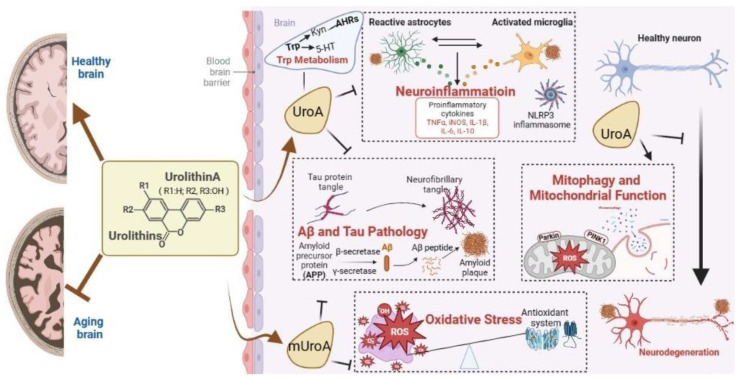
Outline of effects of UroA on brain aging (actions and mechanisms). The mechanisms of UroA on aging brain mainly focus on promoting mitophagy and mitochondrial function and relieving neuroinflammation. Other mechanisms include mitigating oxidative stress, inhibiting Aβ and tau pathology, and regulating Trp metabolism. Although UroA has been shown to activate anti-aging signaling pathways such as AMPK, SIRTs, and mTOR in vivo, the direct or key targets of UroA have not been fully elucidated.

**Table 4 nutrients-15-03884-t004:** IC_50_ of Uros and corresponding references on synthetic and oxygen free radicals.

Sample	Superoxide Radical ^1^	DPPH Radical	Peroxyl Radicals ^2^	ABTS Radical	Hydroxyl Radical
UroA	5.01 ± 5.01 μM	152.66 ± 35.01 μM	13.1 μM	---	---
Gallic acid	0.26 ± 0.21 μM	3.10 ± 3.11 μM	---	---	---
Ascorbic acid	---	14.81 ± 14.90 μM	---	---	---
Pomegranate extracts	---	---	0.49 μM	---	---
UroB	495.32 ± 3.28 mM	295.41 ± 2.36 mM	---	316.18 ± 1.85 mM	306.28 ± 4.61 mM
Ascorbic acid	874.39 ± 1.48 mM	446.25 ± 1.78 mM	---	526.24 ± 3.18 mM	540.16 ± 2.52 mM

IC_50_ is expressed as mean ± SD, and data for peroxyl radicals are expressed as µmol Trolox equivalents per mg of sample. ^1^ UroA scavenges superoxide radicals generated by the xanthine/xanthine oxidase system. ^2^ The capacity of UroA to scavenge peroxyl radicals was measured by the oxygen radical antioxidant capacity (ORAC).

**Table 5 nutrients-15-03884-t005:** Summary of Uros metabolites identified in the brain.

Animals	Route of Administration	Treatment	Brain Tissue	Identified Metabolites in Brain	Plasma Concentratio	Refs
male C57BL/6 mice (7 months, 25–30 g)	i.g.	UroA 0.3 mg/mouse, single administration	brain tissues	mUroA: 8 ng/g	---	[147]
male C57BL/6 mice (6 weeks)	i.g.	UroA 200 mg/kg b.w., single administration	cortex	UroA: 28 ng/g	15 ng/mL	[148]
			hippocampus	UroA: 35 ng/g		
male rats (12 weeks, 288 ± 20 g)	i.v.	Polyphenol metabolites (12.5 μg UroA + 5.3 μgUroB) 2.7 µmol/rat/day for 2 days	brain tissues	UroA: 2.2 ng/g	---	[41]
				UroB: 0.5 ng/g		
male albino Wistar rats (6 weeks, 250–300 g)	i.g.	Pomegranate juice 500 mg/kg b.w./day for 10 days	brain tissues	UroA: 1.68 ± 0.25 ng/g	18.75 ± 3.21 ng/mL	[36]
male albino Wistar rats (6 weeks, 250–300 g)	i.g.	Pomegranate juice 500 mg/kg b.w./day for 45 days	brain tissues	UroA: 2.068 ± 0.274 ng/g	---	[149]

Abbreviations: i.v.: intravenous, i.g.: intragastrical, b.w.: body weight.

## Data Availability

No new data were created.

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
