# Peer review of "Urolithins: A Prospective Alternative against Brain Aging"

_nutrients, 2023, doi:10.3390/nu15183884_

Round 1

Reviewer 1 Report

Congratulations to the authors who wrote a comprehensive and detailed review article in which all previous basic and clinical studies related to urolithins were presented and the pathomechanism of action was explained.

Considering that the mentioned clinical studies include subjects aged 50 years and above and different long periods taking urolithin precursors, what would be the author's conclusion regarding the recommendations for starting to take and what amount of these nutrients to reduce the effect of brain aging and the development of neurodegenerative diseases in general.

Major Comments:

References are not ordered or written according to journal instructions. A space is missing before the reference within the text.

The introduction is vaguely written, so it is difficult to follow because the sentences are too long: lines 32-40. I think that should be shortened or split into two or more sentences.

Minor comments

Line 545 is written: amimals - it should be written animals.

Table 3 reference (70) is missing the arrows in the penultimate column area that explain what has changed. It should also be inserted in Table 3 on page 9 inside the penultimate column on individual findings of the arrow so that it is clear whether there has been an increase or decrease in some variables (it is not uniform in all places although it is assumed that everything is under the same).

Table 2 lacks explanations of some abbreviations (eg STZ, SOD).

Table 3 lacks explanations of some abbreviations (e.g. EPA, DHA).

In Table 4 column 2/title - it is written: radicala - should be written radical.

Reviewer 2 Report

In An et al., the authors provide a comprehensive review of Urolithins (Uros) in age-related brain dysfunctions and neurodegenerative diseases and discuss their prospects as functional food/drugs or biomarkers against brain aging. The authors delineate the positive effects of Uros in brain disorders and various preclinical (Table 1 & 2)/clinical studies (Table 3) to show the beneficial effects of Uros age-related brain dysfunctions. The review also lists the knowledge gaps in the use of Uros as functional foods/drugs/biomarkers. While the review is well organized, with various tables/figures depicting the role and importance of Uros in brain aging and neurodegenerative diseases, with emphasis on their therapeutic targets and mechanisms, and discuss their prospects, there are minor issues that make the quality of the review difficult to assess:

·         A table listing the use of Uros in various neurological disorders would be helpful to the reader.

·         Are any of the Uros extract in clinical trials currently? If so, which phase of clinical trials they are in?
